# Rational Design of Field-Effect Sensors Using Partial Differential Equations, Bayesian Inversion, and Artificial Neural Networks

**DOI:** 10.3390/s22134785

**Published:** 2022-06-24

**Authors:** Amirreza Khodadadian, Maryam Parvizi, Mohammad Teshnehlab, Clemens Heitzinger

**Affiliations:** 1Institute of Applied Mathematics, Leibniz University Hannover, Welfengarten 1, 30167 Hannover, Germany; parvizi@ifam.uni-hannover.de; 2Cluster of Excellence PhoenixD (Photonics, Optics, and Engineering-Innovation Across Disciplines), Leibniz University Hannover, 30167 Hannover, Germany; 3Faculty of Electrical Engineering, K. N. Toosi University of Technology, Tehran 19697, Iran; teshnehlab@eetd.kntu.ac.ir; 4Institute of Analysis and Scientific Computing, TU Wien, Wiedner Hauptstrasse 8–10, 1040 Vienna, Austria; clemens.heitzinger@tuwien.ac.at; 5Center for Artificial Intelligence and Machine Learning (CAIML), TU Wien, 1040 Vienna, Austria

**Keywords:** field-effect sensors, biosensors, charge transport, neural networks, Bayesian inversion, inverse modeling

## Abstract

Silicon nanowire field-effect transistors are promising devices used to detect minute amounts of different biological species. We introduce the theoretical and computational aspects of forward and backward modeling of biosensitive sensors. Firstly, we introduce a forward system of partial differential equations to model the electrical behavior, and secondly, a backward Bayesian Markov-chain Monte-Carlo method is used to identify the unknown parameters such as the concentration of target molecules. Furthermore, we introduce a machine learning algorithm according to multilayer feed-forward neural networks. The trained model makes it possible to predict the sensor behavior based on the given parameters.

## 1. Introduction

Silicon nanowire (SiNW) field-effect transistors (FETs) are typically used to detect proteins [1], cancer cells [2], DNA and miRNA strands [3,4], enzymes [5], and toxic gases such as carbon monoxide [6,7]. The sensors have several advantages including fast response, very high sensitivity, and low power consumption; they do not need labeling and can be used to detect subpicomolar concentrations of biological species [8,9,10,11,12,13]. The functioning of the sensors is based on the field effect due to the (partial) charges of the target molecules. When they are selectively bound to probe molecules and close enough to the semiconducting transducer, they affect the charge concentration inside the nanowire, which changes the current through the nanowire.

Using mathematical models based on partial differential equations (PDEs) enables us to model physically relevant quantities such as electrostatic potential, electron and hole current density, device sensitivity to the target molecule and signal-to-noise ratio [14,15,16,17,18]. The three-dimensional simulations give rise to more reliable models compared to two-dimensional cross-sections, since all target molecules bound to bio-receptors will be included [19,20]. We couple a charge transport model (the drift-diffusion equations) and the nonlinear Poisson–Boltzmann equation (PBE) for fully self-consistent simulations. The system of equations is a comprehensive model to compute the electrical current and study the nonlinear effects of different semiconductor parameters (e.g., doping concentration) and device parameters such as nanowire type (radial, trapezoidal, radial, or rectangular), its dimensions, contact voltages, and insulator thickness on device performance (output and sensitivity).

Having an accurate model enables the rational design of field-effect sensors. However, in the model equations, there are several material parameters that cannot be (easily) measured. The surface charge density of the insulator has an essential effect on the device and also affects the probe and target molecules. The doping concentration has a crucial effect on the device and the model. Due to the nonlinear effect of these parameters, an efficient parameter estimation framework will enhance the accuracy and reliability of the model.

Markov-chain Monte-Carlo (MCMC) techniques are among the most efficient probabilistic methods to extract information by comparison between measurements and simulations by updating available prior knowledge and estimating the posterior densities of unknown quantities of interest. Here, we use a forward model, and a backward, inverse setting is used to determined the unknown parameters using the experiments. The classical algorithm was introduced in 1970 and is called the *Metropolis–Hastings (MH) algorithm* [21]. There are several improvements in the algorithm, e.g., adaptive-proposal Metropolis [22], delayed-rejection Metropolis [23], and delayed rejection adaptive Metropolis (DRAM) [24], as well as using ensemble Kalman filters [25]. In all techniques, different candidates are proposed based on a proposal distribution, and the algorithm decides whether they are rejected or accepted. A review of the MCMC methods is given in [26]. For SiNW-FETs, the DRAM algorithm has been used to identify the doping concentration and the amount of target molecules [14]. Considering the selective functionalization of SiNW, the authors of [1] used the MH algorithm to estimate the probe-target density at the surface.

Neural networks (also known as artificial neural networks (ANNs)) as the subset of machine learning are frameworks to analyze the available data and discover patterns that can not be observed independently. The ANNs have been inspired by the human brain and are suitable for complicated and nonlinear cases. Here, we split the prior data into two categories, namely training and testing data. The training set (between 60% and 80%) is used to extract useful information from the data, and the test set (between 20% and 40%) is employed to monitor the algorithm performance. In SiNW-FETs, there are a large amount of simulation and experimental data concerning different input (physical, chemical, and device) parameters that should be analyzed to ensure their accuracy and reliability. Of course, this process is time consuming and reduces the efficiency. Furthermore, the sensors are developed to detect specific biological species with the highest sensitivity. In the design process, using neural networks enables us to optimize the design parameters to enhance the sensor performance [27,28,29,30,31,32].

This article is structured as follows. In Section 2, we present the model equations and explain how the electrical current is computed. In Section 3, we discuss the parameter estimation methods and explain how MCMC can be used to determine the unknown parameters. In Section 4, we introduce the developed neural networks algorithm for SiNW-FETs. In Section 5, we first verify the model response with the experimental data; then, Bayesian inversion is used to identify the material parameters. Afterward, the developed machine-learning algorithm is employed in training and testing. Finally, the conclusions are summarized in Section 6.

## 2. The Model Equations

The drift–diffusion–Poisson system is used to describe the electrochemical interactions (Poisson–Boltzmann equation) and the charge transport (drift–diffusion equations) in field-effect sensors. The convex and bounded domain $\Omega \subset R^{3}$ consists of four subdomains, namely the insulator (SiO${}_{2}$, $\Omega_{\mathrm{Si}}$), the silicon substrate and transducer ($\Omega_{\mathrm{Si}}$), the aqueous solution ($\Omega_{\mathrm{liq}}$), and the charged molecules ($\Omega_{\mathrm{mol}}$). To model the potential interactions, we use the Poisson–Boltzmann equation
(1)$$\begin{array}{cc} -\nabla \cdot (A(x)\nabla V(x))\; & =\left\{\begin{array}{cc} q(C_{\mathrm{dop}}\left(x\right)+p\left(x\right)-n\left(x\right)) & \mathrm{in}\;\Omega_{\mathrm{Si}},\\ 0 & \mathrm{in}\;\Omega_{\mathrm{ox}},\\ \rho (x) & \mathrm{in}\;\Omega_{M},\\ -2\varphi \left(x\right)\sinh(\beta \left(V\left(x\right)-\Phi_{F})\right) & \mathrm{in}\;\Omega_{\mathrm{liq}}, \end{array}\right. \end{array}$$
where *A* indicates the dielectric constant, which is a function of the material, *V* is the electrostatic potential, $C_{\mathrm{dop}}$ is the doping concentration, ρ is the surface charge of the molecules, $\Phi_{F}$ denotes the Fermi level, and φ is the ionic concentration. Regarding the electrical constants, we use the relative values $A_{\mathrm{Si}}=11.7$, $A_{\mathrm{ox}}=3.9$, $A_{M}=3.7$, and $A_{\mathrm{liq}}=78.4$. Considering the Boltzmann constant $k_{B}$, the temperature *T* and the elementary charge *q*, we define $\beta =q/(k_{B}\;T)$. In the simulations, a thermal voltage of 0.021V will be used.

A two-dimensional cross-section of the device is given in Figure 1.

At the interface between the insulator and the liquid (i.e., $\Gamma :=\Omega_{\mathrm{ox}}\cap \Omega_{\mathrm{liq}}$), we impose the interface conditions
(2a)$$\begin{array}{ccc} A\left(0+\right)\left(V(0+,y,z)-V(0-,y,z)\right) & =\alpha (y,z)\; & \mathrm{on}\;\Gamma , \end{array}$$
(2b)$$\begin{array}{ccc} A\left(0+\right)\partial_{x}V\left(0+,y,z\right)-A\left(0-\right)\partial_{x}V\left(0-,y,z\right) & =\gamma (y,z)\; & \mathrm{on}\;\Gamma \end{array}$$
for $V_{I}$. Here, 0+ and 0− denote the limit at the interface on the side of liquid and insulator. Furthermore, α is macroscopic dipole moment density, and γ is the macroscopic surface-charge density.

In $\Omega_{\mathrm{Si}}$, we solve the drift–diffusion system
(3a)$$\begin{array}{cc} -\nabla \cdot (A\nabla V) & =q(p\left(x\right)-n\left(x\right)+C_{\mathrm{dop}}\left(x\right)), \end{array}$$
(3b)$$\begin{array}{cc} \nabla \cdot J_{n} & =qR(n,p), \end{array}$$
(3c)$$\begin{array}{cc} \nabla \cdot J_{p} & =-qR(n,p), \end{array}$$
(3d)$$\begin{array}{cc} J_{n} & =q(D_{n}\nabla n-\mu_{n}n\nabla V), \end{array}$$
(3e)$$\begin{array}{cc} J_{p} & =q(-D_{p}\nabla p-\mu_{p}p\nabla V) \end{array}$$
to model the charges in the transistor, where $D_{n}$ and $D_{p}$ are the electron and hole diffusion coefficients. The concentrations of electrons and holes are given by
(4)$$p=:n_{i}\exp\left(\frac{q}{K_{B}T}\left(\Phi_{1}-V\right)\right),\;n=:n_{i}\exp\left(\frac{-q}{K_{B}T}\left(\Phi_{2}-V\right)\right),$$
where $n_{i}$ is the intrinsic carrier density and $\Phi_{1}$ and $\Phi_{1}$ are the Fermi levels. In order to compute the electron and hole current densities, we use the Shockley–Read–Hall recombination rate, i.e.,
$$R\left(n,p\right):=\frac{np-n_{i}^{2}}{\tau_{n}\left(p+n_{i}\right)+\tau_{p}\left(n+n_{i}\right)},$$
where $\tau_{n}$ and $\tau_{p}$ denote the lifetimes of the electrons and holes.

For solving the nonlinear system of equations, we use the Scharfetter–Gummel iteration. For this, we write the concentrations *n* and *p* in terms of the two Slotboom variables *u* and *v* as
(5a)$$\begin{array}{cc} n(x,\omega ) & =:n_{i}e^{V\left(x,\omega \right)/U_{T}}u\left(x,\omega \right), \end{array}$$
(5b)$$\begin{array}{cc} p(x,\omega ) & =:n_{i}e^{-V\left(x,\omega \right)/U_{T}}v\left(x,\omega \right). \end{array}$$
Therefore, the model problem (3) can be rewritten as
(6a)$$\begin{array}{cc} -\nabla \cdot (A(x)\nabla V(x)) & =q\left(C_{\mathrm{dop}}\left(x\right)-n_{i}\left(e^{V\left(x\right)/U_{T}}u\left(x\right)-e^{-V\left(x\right)/U_{T}}v\left(x\right)\right)\right), \end{array}$$
(6b)$$\begin{array}{cc} U_{T}n_{i}\nabla \cdot \left(\mu_{n}e^{V/U_{T}}\nabla u\left(x\right)\right) & =R(x), \end{array}$$
(6c)$$\begin{array}{cc} U_{T}n_{i}\nabla \cdot \left(\mu_{p}e^{-V/U_{T}}\nabla v\left(x\right)\right) & =R(x), \end{array}$$
where $U_{T}$ is the thermal voltage and the Shockley–Read–Hall recombination rate takes the form
$$R_{\mathrm{SRH}}\left(x\right)=n_{i}\frac{u(x)v(x)-1}{\tau_{p}\left(e^{V/U_{T}}u\left(x\right)+1\right)+\tau_{n}\left(e^{-V/U_{T}}v\left(x\right)+1\right)}.$$

At the ohmic contacts (backgate, source, and drain) and the solution gate, we have a Dirichlet boundary condition $V_{\partial \Omega }=V_{D}$ consisting of
(7)$$\begin{array}{c} {V|}_{\partial \Omega_{G}}=V_{g}{\;V|}_{\partial \Omega_{S}}=V_{S}\;V{{|}_{\partial \Omega_{D}}=V_{D}\;V|}_{\partial \Omega_{\mathrm{sol}}}=V_{\mathrm{solution}}. \end{array}$$
At the source and drain contacts ($\mathrm{on}\;\partial \Omega_{\mathrm{Si}}$), we apply
(8)$$\begin{array}{c} u\left(x\right)=u_{D}\left(x\right),\;v\left(x\right)=v_{D}\left(x\right). \end{array}$$
For the remaining part of the domain, we impose a zero Neumann boundary condition to guarantee the self-isolation. We refer the interested reader to [15,19,33,34] for theoretical discussions about the model including the Slotboom variables. The existence and uniqueness of the solutions for deterministic and stochastic model problems are given in [15,35]. Finally, the computation of $J_{n}$ and $J_{p}$ enables us to calculate the electrical current as
(9)$$\begin{array}{c} I:=\int \left(J_{n}+J_{p}\right)\;dx, \end{array}$$
where we take the integral on a cross-section of the transducing part.

In this work, we use the finite element method (FEM) to solve the coupled system of equations. We define the spaces
(10a)$$\begin{array}{cc} X^{1} & =\left\{V\in H^{1}{\left(\Omega \right)\;\;\;|\;V|}_{\partial \Omega }=V_{D}{\;,\;\;V|}_{\Gamma }=V_{I}\right\}, \end{array}$$
(10b)$$\begin{array}{cc} X^{2} & =\left\{u\in H^{1}\left(\Omega_{\mathrm{Si}}\right)\;{\left|\;u\right|}_{\partial \Omega_{\mathrm{Si}}}=u_{D}\right\}, \end{array}$$
(10c)$$\begin{array}{cc} X^{3} & =\left\{v\in H^{1}\left(\Omega_{\mathrm{Si}}\right)\;{\left|\;v\right|}_{\partial \Omega_{\mathrm{Si}}}=v_{D}\right\}. \end{array}$$
Therefore, we define the continuous solution space $X:=X^{1}\times X^{2}\times X^{3}$ for the DDP system. Regarding the space discretization, we assume $T_{h}=\left\{T_{1},T_{2},\ldots ,T_{n}\right\}$ denotes a quasi-uniform mesh defined on $\Omega_{h}\approx \Omega$ with mesh width $h:=\max_{T_{j}\in T_{h}}\mathrm{diam}\left(T_{j}\right)$. We define
$$\begin{array}{cc} S_{V}^{1}\left(T_{h}\right) & :=\{V\in H^{1}\left(\Omega \right)\;|\;V{|}_{T}\in P_{1}\left(T\right)\;\;\forall T\in T_{h}\},\\ S_{u}^{1}\left(T_{h}\right) & :=\{u\in H^{1}\left(\Omega \right)\;\;|\;u{|}_{T}\in P_{1}\left(T\right)\;\;\forall T\in T_{h}\},\\ S_{v}^{1}\left(T_{h}\right) & :=\{v\in H^{1}\left(\Omega \right)\;\;|\;v{|}_{T}\in P_{1}\left(T\right)\;\;\forall T\in T_{h}\}, \end{array}$$
where $P_{1}$ is the space of first-order polynomials. Then, we have
(11a)$$\begin{array}{cc} X_{h}^{1} & :=\left\{V_{h}\in S_{V}^{1}\left(T_{h}\right)\;\;\;|\;V_{h}{{|}_{\partial \Omega }=V_{D}\;,\;\;V_{h}|}_{\Gamma }=V_{I}\right\}, \end{array}$$
(11b)$$\begin{array}{cc} X_{h}^{2} & :=\left\{u_{h}\in S_{u}^{1}\left(T_{h}\right)\;\;\;{\left|\;u_{h}\right|}_{\partial \Omega_{\mathrm{Si}}}=u_{D}\right\}, \end{array}$$
(11c)$$\begin{array}{cc} X_{h}^{3} & :=\left\{v_{h}\in S_{v}^{1}\left(T_{h}\right)\;\;\;{\left|\;v_{h}\right|}_{\partial \Omega_{\mathrm{Si}}}=v_{D}\right\}. \end{array}$$
The discrete solution is defined as $X_{h}:=X_{h}^{1}\times X_{h}^{2}\times X_{h}^{3}$, which is a subset of *X*. The weak form of the model equations can be found in [15,33]. The a prior and a posterior estimations are proved in [33]. More theoretical works regarding the finite elements analysis are given in [36,37,38].

## 3. Parameter Estimation Based on Bayesian Inference

In different experimental situations, an accurate estimation of the effective parameters and constants cannot be easily estimated. Bayesian inversion techniques based on Markov chain Monte Carlo methods are efficient and straightforward probabilistic techniques to estimate these unknowns. We initiate the algorithm using the available information, named *prior* knowledge (which may not be sufficiently accurate), and during several iterations, we can update the information and provide more reliable data (i.e., the posterior density). Then, we can extract valuable information from the posterior density, and its mean/median can be used as the solution of the interference. A very strong agreement with the experimental values and the model response can be achieved. We start a statistical model
(12)$$\begin{array}{c} M=P(x,\chi )+\varepsilon , \end{array}$$
where M is the experimental observation (normally n− dimensional), while P is the solution of the model problem which depends on the set of parameters χ (i.e., $\chi =\left(\chi_{1},\chi_{2},\ldots ,\chi_{k}\right)$ and the Cartesian coordinates *x*. Here, ε is the measurement error, and we assume that it is normally distributed, i.e., $\varepsilon ∼N(0,\sigma^{2}\;I)$, including the parameter $\sigma^{2}$. Having an experimental observation, for instance electrical current (i.e., M=obs), we define the probability function
(13)$$\begin{array}{c} \pi \left(\mathrm{obs}\right)=\int_{R^{n}}\pi \left(\mathrm{obs}|\chi \right)\pi_{0}\left(\chi \right)\;d\chi . \end{array}$$
Our aim is to estimate the posterior density π(χ|m), considering the measured observation *m* and the available prior information. For this, we compute the likelihood function
(14)$$\begin{array}{c} \pi \left(M|\chi \right)=L\left(\chi ,\sigma^{2}|M\right)=\frac{1}{{\left(2\pi \sigma^{2}\right)}^{n/2}}\exp\left(-M_{P}/2\sigma^{2}\right) \end{array}$$
where
(15)$$\begin{array}{c} M_{P}=\sum_{j=1}^{n}{\left[M_{j}-P_{j}\left(x,\chi \right)\right]}^{2} \end{array}$$
is the sum of square errors. Obviously, if the model response with respect to the (set of) parameters χ will be closer to the measured value, the square error (Equation 15) will converge to zero, and its relative probability (computed by the likelihood function) will converge to 1. Inaccurate estimation of χ will increase the error term, and the probability will converge to zero.

In the Metropolis algorithm, we initiate the process using an initial guess $\chi^{0}$ based on the prior density. According to the proposal distribution, a new candidate $\chi^{🟉}$ is proposed. We compute the acceptance rate by
(16)$$\begin{array}{c} \lambda \left(\chi^{j-1},\chi^{🟉}\right)=\min\left(1,\frac{\pi (\chi^{🟉})}{\pi (\chi^{j-1})}\right). \end{array}$$
If the new candidate $\chi^{🟉}$ is accepted, we continue the MCMC chain with that; otherwise, ($\chi^{j-1}$ has a higher probability concerning $\chi^{🟉}$), we follow the chain with the previous candidate. Using a non-symmetric proposal density is a generalization of the Metropolis algorithm, introduced by Hastings [21], where the probability of the forward jump is not equal to the backward one. A summary of the algorithm is given in Algorithm 1.
**Algorithm 1** The Metropolis–Hastings algorithm.Initialization: Start the process with the initial guess $\chi^{0}$ and number of samples *N*.**while**j<N   1. Propose a new sample according to the proposal density $\chi^{*}∼T\left(\chi^{*}|\;\chi^{j-1}\right)$.   2. Compute the acceptance/rejection ratio
$$\zeta \left(\chi^{*}|\;\theta^{j-1}\right)=\min\left(1,\frac{\pi (\chi^{*}|\;m)}{\pi (\chi^{j-1}|\;m)}\;\frac{T(\chi^{j-1}|\;\chi^{*})}{T(\chi^{*}|\;\chi^{j-1})}\right).$$   3. Sample R∼Uniform(0,1).   4.ifR<ζ **then**         accept $\chi^{*}$ and set $\chi^{j}:=\chi^{*}$      **else**         reject $\chi^{*}$ and set $\chi^{j}:=\chi^{j-1}$      **end if**   5. Set j=j+1.

The Metropolis–Hastings algorithm is a simple and versatile technique and has been widely used for several problems in applied science. However, for the high-dimensional cases (different parameters should be inferred simultaneously), the algorithm does not work appropriately, since the rejection rate increases significantly. To improve its computational drawbacks, different improvements, such as the adaptive Metropolis algorithm [22], delayed rejection Metropolis [23], and their combination, namely delayed rejection adaptive Metropolis (DRAM) [24]. We refer the interested readers to [26] as a review paper about the methods.

### Mcmc with Ensemble-Kalman Filter (EnKF-MCMC)

In EnKF-MCM [25], we use a Kalman gain employing the mean and the covariance of the prior distribution and the cross-covariance between parameters and observations. It will be used to compute the proposal distribution and make the convergence to the target density faster. Here, the new candidate is computed as the jump of the Kalman-inspired proposal Δχ as
(17)$$\begin{array}{c} \chi^{🟉}=\theta^{j-1}+\Delta \chi . \end{array}$$
In order to update the candidates, we compute Δχ by
(18)$$\begin{array}{c} \Delta \chi =K\left(y^{j-1}+s^{j-1}\right), \end{array}$$
where K denotes the so-called Kalman gain,
(19)$$\begin{array}{c} K=C_{\chi M}{\left(C_{MM}+R\right)}^{-1}. \end{array}$$
Here, $C_{\theta M}$ indicates the covariance matrix between the identified unknowns and model response, $C_{MM}$ points out the covariance matrix of the model response, and R denotes the measurement noise covariance matrix [39]. In addition, $y^{j-1}$ is the residual of the proposed values concerning the model and $s^{j-1}∼N\left(0,R\right)$ relates to the density of measurement. A summary of the relative algorithm is given in Algorithm 2. Finally, Figure 2 shows the implementation of EnKF-MCMC and Schafetter–Gummel iteration for parameter estimation and solving the model equations.
**Algorithm 2** Bayesian inference using EnKF-MCMC**Initialization (j=0)**: Start the process with the initial guess $\chi^{0}$ and number of samples *N*.**while**j<N      1. Estimate the model response with respect to $\chi^{j-1}$      2. Compute the Kalman gain    $K=C_{\chi M}{\left(C_{MM}+R\right)}^{-1}$      3. Produce the new proposal using the shift    $\chi^{🟉}=\chi^{j-1}+K\left(y^{j-1}+s^{j-1}\right)$      4. Accepted/rejected $\chi^{🟉}$      5. Set j=j+1.


## 4. Multilayer Feed-Forward Neural Networks

Neural networks are efficient, flexible, and robust simulation tools specifically for nonlinear and complicated problems. They consist of three effective components, including neurons, structures, and weights, which all affect the response and behavior of the network. Artificial neural networks (ANNs) are supervised machine learning algorithms consisting of neurons and hidden layers. The input data are processed into the hidden layers, the output is compared with the target trajectory, and the relative error is computed. The neural networks strive to minimize this error.

Typically, there are two common classes of neural networks, namely feed-forward neural networks (single or multilayers) and recurrent dynamics neural networks. Single-layer neural networks [40] have less complexity; however, they are more suitable for linear problems. In multilayer feed-forward neural networks (MFNNs) [41,42], more than one layer of the artificial neurons will be used to enhance the capability to learn nonlinear patterns, which is more appropriate for BIO-FETs. In MFNNs, the neurons are organized in different non-recurrent layers, where in the first layer, we have the input vector (here are the parameters of the sensor), and the output is given to the first hidden layer. After the data processing, the data are transferred to the next layers using the weights; the procedure is followed until the latest MFNNs layer. These networks are also named multilayer perceptrons, and their structure is shown in Figure 3.

Let us assume *d* denotes the desired trajectory (i.e., the device output); for *M*-layer neural networks, we have
$$\begin{array}{cc} \underset{(n_{s-1})\times 1}{\nabla w_{j}^{s}\left(k\right)} & =\eta^{s}\frac{\partial E}{\partial w_{j}^{s}}\left(k\right)=-\eta^{s}\delta_{i}^{s}\left(k\right)\underset{(n_{s-1})\times 1}{\nabla w_{j}^{s}\left(k\right)} \end{array}$$
(20)$$\begin{array}{cc} \; & =-\eta^{s}e_{j}^{s}\left(k\right)f_{j}^{s^{\prime }}\left(net_{j}^{s}\left(k\right)\right)\underset{(n_{s-1})\times 1}{x^{s-1}}\;s=1,2,\ldots ,M,\;j=1,2,\ldots ,n_{s}, \end{array}$$
(21)$$\begin{array}{cc} \delta_{j}^{s}\left(k\right) & :=-\frac{\partial E}{\partial net_{j}^{s}}\left(k\right)=e_{j}^{s}\left(k\right)f_{j}^{s^{\prime }}\left(net_{j}^{s}\left(k\right)\right), \end{array}$$
(22)$$\begin{array}{cc} e_{j}^{s}\left(k\right) & =\sum_{l=1}^{n_{s+1}}\delta_{j}^{s+1}\left(k\right)w_{lj}^{s+1}\left(k\right), \end{array}$$
where *w* is the weights, η is the training rate, *E* is the network mean square error (MSE), δ is the sensitivity function (here, $\delta^{s}$ indicates the network error in the *j*th layer), $net^{s}$ is the weighted input, $n_{s}$ is the number of neurons in the *s*th layer, $x^{0}$ is the network input, $x^{s-1}$ is the output of the s−1th layer, and it is also the input of the *s*th layer. We also have the following initial conditions for the recurrent process
(23)$$\begin{array}{cc} \delta_{j}^{M}\left(k\right) & =e_{j}^{M}\left(k\right)f_{j}^{M^{\prime }}\left(net_{j}^{M}\left(k\right)\right), \end{array}$$
(24)$$\begin{array}{cc} e_{j}^{M}\left(k\right)) & ≜d_{j}\left(k\right)-O_{j}^{M}\left(k\right). \end{array}$$
Figure 4 shows the *j*th neuron in the *i*th layer in the learning algorithm. In the recurrent process, in order to adjust the weights from the first layer, we follow as
(25)$$\begin{array}{cc} \delta_{l_{i}}^{s}\left(k\right) & =-\frac{\partial E(k)}{\partial net_{l_{i}}^{s}}\sum_{l_{m}=1}^{n_{M}}\sum_{l_{m-1}=1}^{n_{M-1}}\cdots \sum_{l_{i+2}=1}^{n_{i+2}}\sum_{l_{i+1}=1}^{n_{i+1}}\frac{\partial E}{\partial net_{l_{m}}^{s}}\frac{\partial net_{l_{m}}^{s}}{\partial net_{l_{m}-1}^{s-1}}\cdots \frac{\partial net_{l_{i+2}}^{s+2}}{\partial net_{l_{i+1}}^{s+1}}\frac{\partial net_{l_{i+1}}^{s+1}}{\partial net_{l_{i}}^{s}}\left(k\right) \end{array}$$

For i=1,2,…,M−1 and s=1,2,…,M, the relation $net_{l_{i}}^{s}$ and $net_{l_{i+1}}^{s+1}$ takes
(26)$$\begin{array}{c} net_{l_{i+1}}^{s+1}\left(k\right)=\sum_{p=1}^{n_{i}}w_{l_{i+1}p}^{s+1}\left(k\right)f_{p}^{s}\left(net_{p}^{s}\left(k\right)\right), \end{array}$$
therefore
(27)$$\begin{array}{c} \frac{\partial net_{l}^{s+1}}{\partial net_{l_{i}}^{s}}\left(k\right)=w_{l_{i+1}l_{i}}^{s+1}\left(k\right)f_{l_{i}}^{s^{\prime }}\left(net_{l_{i}}^{s^{\prime }}\left(k\right)\right). \end{array}$$
So, we can write $\delta_{l_{i}}^{s}$ as
(28)$$\begin{array}{c} \delta_{l_{i}}^{s}\left(k\right)=\left(\sum_{l=1}^{n_{i+1}}\delta_{l}^{s+1}\left(k\right)w_{l\;l_{i}}^{s+1}\left(k\right)\right)f_{l_{i}}^{s^{\prime }}\left(net_{l_{i}}^{s}\left(k\right)\right)=e_{l_{i}}^{s}\left(k\right)f_{l_{i}}^{s^{\prime }}\left(net_{l_{i}}^{s}\left(k\right)\right), \end{array}$$
where
(29)$$\begin{array}{c} e_{l_{i}}^{s}\left(k\right)=\sum_{l=1}^{n_{i+1}}\delta_{l}^{s+1}\left(k\right)w_{l\;l_{i}}^{s+1}\left(k\right). \end{array}$$
The gradient of *E* (the difference between desired trajectory and the neural networks’s output) with respect to the weight vector is given by
(30)$$\begin{array}{c} \frac{\partial E}{\partial w_{l_{i}}^{s}}\left(k\right)=\sum_{l=1}^{n}\frac{\partial E}{\partial net_{l_{i}}^{s}}\left(k\right)\frac{\partial net_{l_{i}}^{s}}{\partial w_{l_{i}}^{s}}\left(k\right), \end{array}$$
where the second term depends only on the neurons features and takes
(31)$$\begin{array}{c} \frac{\partial net_{l}^{s}}{\partial w_{l_{i}}^{s}}\left(k\right)=\left\{\begin{array}{cc} x^{s-1}\left(k\right)\;\; & \;l=l_{i},\\ 0 & \mathrm{otherwise}, \end{array}\right. \end{array}$$
(32)$$\begin{array}{c} \frac{\partial E}{\partial w_{l_{i}}^{s}}\left(k\right)=-\delta_{l_{i}}^{s}\left(k\right)x^{s-1}\left(k\right). \end{array}$$
Using the back-propagation error algorithm enables us to adjust the weight functions in order to minimize the network error. This training process is also named the supervised learning algorithm.

## 5. Numerical Experiments

As we already mentioned, the DDP system is a roust and reliable system of equations to model the electrical behavior of the FET devices. We use a prostate-specific antigen (PSA) sensitive sensor which is used to diagnose prostate cancer. For the simulations, we use a sensor device with the nanowire length of 1000 nm, width of 100 nm and height of 50 nm, which is coated with SiO${}_{2}$ with 8 nm thickness. We use the $P_{1}$ finite element to solve the model problem, and tetrahedral meshes are employed to discretize the domain. A schematic of the bio-FET including dimensions using 6622 nodes and 45,735 tetrahedra is shown in Figure 5. The sensor is developed for the detection of 2ZCH (https://www.rcsb.org/structure/2ZCH). The PROPKA algorithm predicts the pK${}_{a}$ values of ionizable groups in proteins and protein–ligand complexes based on the 3D structure. The values are the basis for understanding the pH-dependent characteristics of proteins and catalytic mechanisms of many enzymes [43]. To compute the net charge, we performed a PROPKA algorithm [44,45,46] to detect the net charge for different pH values. The simulations are completed using a pH value of 9, giving rise to the net charge of −15 q [14]. In field-effect sensors, surface reactions at the oxide surface depending on the pH value and the binding of charged target molecules result in changes in the charge concentration at and near the surface, and subsequently in changes in the electrostatic potential, which then modulates the current through the transducer. Since the molecules are negatively charged, the binding of the target molecules to the bio-receptors will enhance the charge conductance and increase the response of the sensor (i.e., the electrical current).

The system of equations is capable of modeling the surface charges at the surface. In a previous work, we developed a Monte-Carlo approach to simulate the charges around a charged biomolecule at a charged surface [47]. Furthermore, in [48], a nonlinear Poisson model was used to calculate the free energies of various molecule orientations in dependence of the surface charge. Based on the free energies, the probabilities of the orientations were calculated, and hence, the biological noise was simulated.

### 5.1. Model Verification

As the first step, we verify the model accuracy with the experiments. We compute the electrical current *I* with respect to different gate voltages $V_{G}$ where the source-to-drain voltage $V_{\mathrm{SD}}=0.2$ V, doping concentration $C_{\mathrm{dop}}=1\times 10^{16}$ cm${}^{-3}$, and the thermal voltage $U_{T}=0.021\;V$. The experimental data are taken form [20]. In order to solve the nonlinear coupled system of equations, a Scharfetter–Gummel-type iteration is used. Figure 6 shows the current as a function gate voltage varying between $V_{G}=-1\;V$ and $V_{G}=-3.5\;V$ for experimental and simulation values. These results indicate that the DDP system is reliable and will be used for the next simulations.

### 5.2. Bayesian Inversion

The molecules are negatively charged (here, −15 q is used); however, an accurate estimation of the molecule charge density will be necessary. In semiconductor devices, in order to enhance the conductivity, impurity atoms are added to the silicon lattice, namely the doping process. Higher doping concentration will improve the transistor conductivity; however, the device will be less sensitive to the charged molecules. Physically, doping concentration (as a macroscopic quantity) denotes the average amount of the dopants. We implemented a delayed rejection adaptive Metropolis (DRAM) [14] and the Metropolis–Hastings algorithm [1] to infer doping concentration, molecule charge density, and probe–target density. The efficiency of the EnKF-MCMC compared to these algorithms is studied in [26]. Therefore, we employ the Kalman filter for the proposal adaptation. We performed the MCMC algorithm with *N* = 10,000 iterations, and a uniform prior density is used. The computational aspects are summarized in Table 1.

The back-propagation error is an efficient algorithm for the training of neural networks where we compute the gradient of the loss function with respect to the weights of the network.

Employing a footprint of 10 nm for the molecules [20,49] gives rise to a surface charge of −1.5 q/nm${}^{2}$. In the experiments, a doping concentration of 1$\times \;10^{16}$ is used in the transducer (both values are selected as the true values). The posterior densities are shown in Figure 7. As expected, the posterior densities are around the true values. Regarding the surface charge, we have a normal distribution, and the charge cannot be positive (which is reasonable due to using P-type FET). For the doping concentration, the distribution points out that for $C_{\mathrm{dop}}$ more than $2\times \;10^{16}$, the sensitivity will reduce significantly, and almost all of the candidates are rejected.

### 5.3. Machine Learning Based on MFNNs

In this section, we employ MFNNs to train the machine according to available information from the sensors. The effective physical/geometrical parameters will have a nonlinear effect on the device output. For instance, for a doping concentration of more than $C_{\mathrm{dop}}=2\times 10^{16}$, the current will increase sharply, which is compatible with the results in Bayesian inversion (Figure 7). Due to this nonlinear behavior, the MFNNs algorithm is chosen to monitor the data accuracy and reliability and predict the sensor behavior.

More hidden layers will facilitate the convergence to the desired trajectory; however, it will increase dramatically the computational costs (e.g., computational time). In this work, we use two hidden layers for the MFNNs algorithm to strike a balance between complexity and efficiency. The procedure is shown in Figure 8. We define five specific scenarios according to the number of inputs. In Case 1, we only have one input (V${}_{g}$) varying between −1V and −5V, where other parameters including insulator thickness, nanowire width (N${}_{W}$), doping concentration, and nanowire height (N${}_{H})$ are constant. In Case 5, we have five inputs, and the output is the calculated electrical current. Table 2 shows the range of the parameters used for different cases.

The MFNNs algorithm is trained with two learning rates (i.e., η=0.1 and η=0.2) and different numbers of epochs. Here, we use 75% of the samples for data training and 25% of the samples for data testing. The numbers of epochs and neurons in the 1st and 2nd hidden layers are given in Table 3. The sigmoid function is used as an activation function in hidden and output layers. In order to verify the efficiency/accuracy of the MFNNs structure algorithm, for different cases, we compare the output of the machine learning algorithm with the desired trajectories (computed currents). We have the relative MSE for the test and training process and performed a linear regression test to explain the relation between the targets and MFNNs output. Figure 9 and Figure 10 show the results for Cases 1–5, where in all cases, there is a good agreement between the machine learning output and the sensor data.

## 6. Conclusions

In this work, we introduced a computational framework for modeling charge transport and electrostatic potential distribution in SiNW-FETs in order to enable the rational design of this sensor technology. The PDE-based model has been verified with the experimental data and showed its accuracy. Bayesian inversion can be used to determine quantities of interest such as molecule concentrations, surface charges, and doping concentrations.

Our approach and results can be extended to different types of sensors including plasma resonance-based biosensors, fluorescence-based sensors, and electrochemiluminescence-based biosensors that are used to detect biomarkers.

Finally, machine learning algorithms based on MFNNs have been developed for SiNW-FETs. Here, we use two hidden layers to deal with the nonlinear behavior of the current (with respect to the input parameters), where the method shows its computational efficiency. We used 75% of the data to train the machine and the remaining 25% for testing. In both cases, the obtained MSE shows the convergence to the desired trajectory. The results indicate that MFNNs are a suitable machine learning algorithm for SiNW-FETs and can be used to predict the sensor output behavior as a compact model.

## Figures and Tables

**Figure 1 sensors-22-04785-f001:**
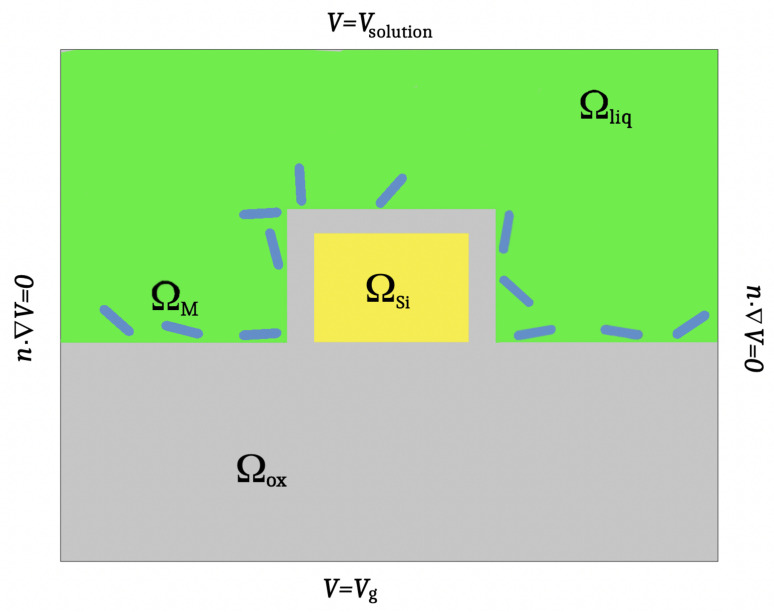
A schematic cross-section of a SiNW-FET depicting the subdomains, i.e., the transducer $\Omega_{\mathrm{Si}}$, SiO${}_{2}$ insulator ($\Omega_{\mathrm{ox}}$), the aqueous solution $\Omega_{\mathrm{liq}}$, the binding of the target molecules to the immobilized receptor molecules ($\Omega_{\mathrm{mol}}$), and the boundary conditions.

**Figure 2 sensors-22-04785-f002:**
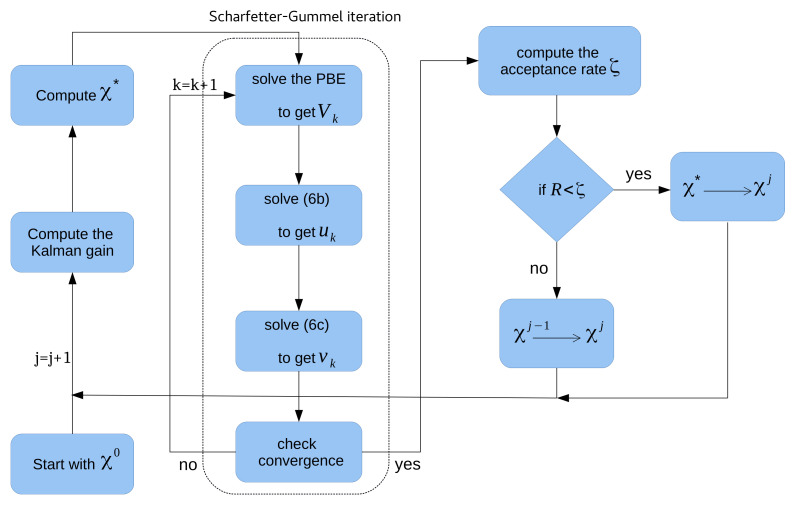
Bayesian inversion using EnKF-MCMC to identify the unknown material parameters, where the Scharfetter–Gummel iteration is used to solved the coupled system of equations.

**Figure 3 sensors-22-04785-f003:**
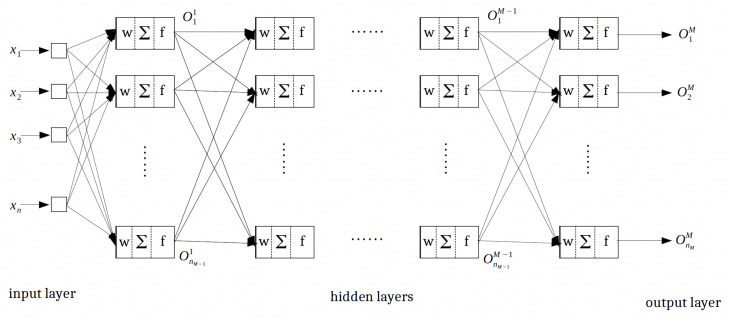
The structure of multilayer feed-forward neural networks (MFNNs).

**Figure 4 sensors-22-04785-f004:**
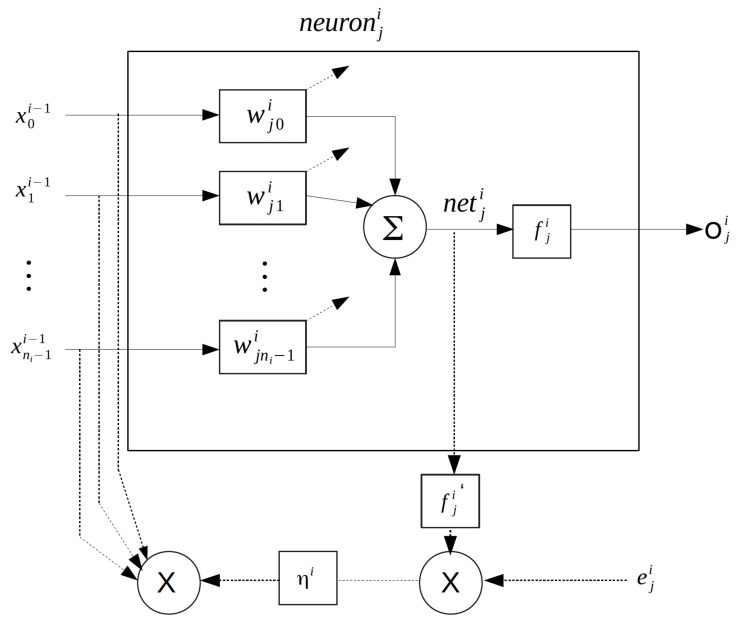
The back-propagation algorithm for the adjustment of neuron weights.

**Figure 5 sensors-22-04785-f005:**
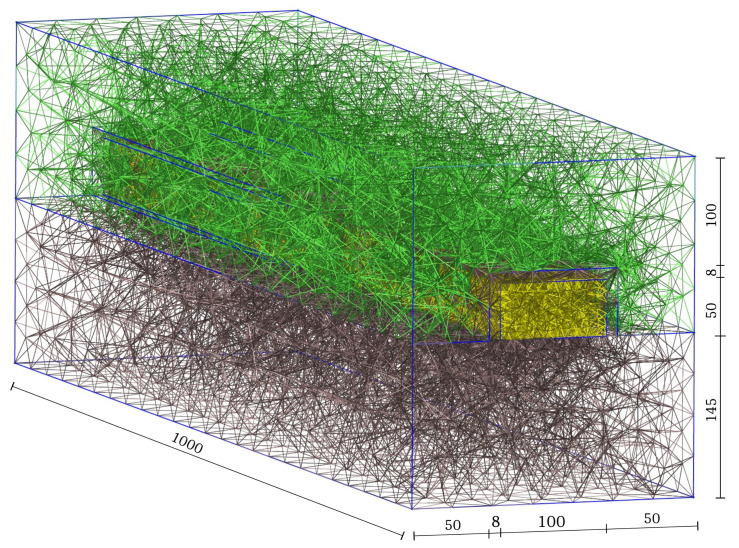
A 3D schematic of the sensor device including the dimensions and tetrahedral meshes for the discretization. All values are in nanometers.

**Figure 6 sensors-22-04785-f006:**
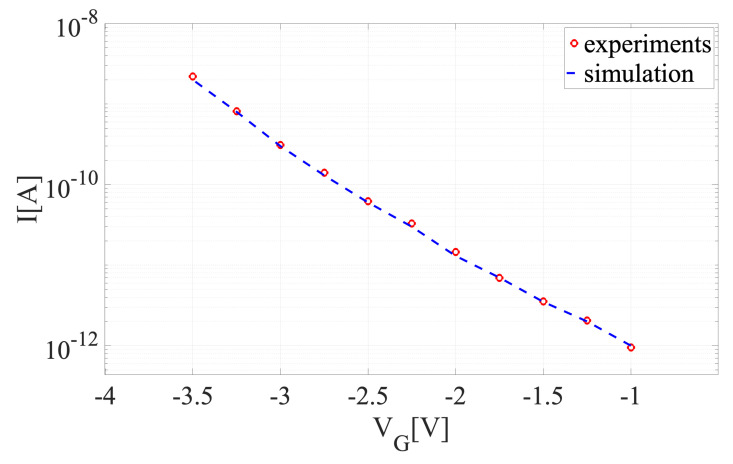
A comparison between the experimental [20] and simulation current.

**Figure 7 sensors-22-04785-f007:**
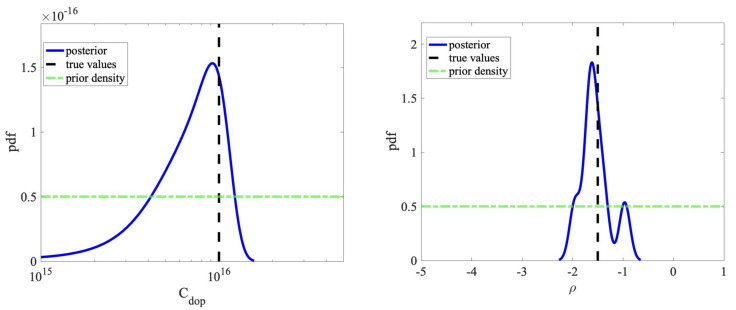
The posterior density of doping concentration (**left**) and surface charge density (**right**) using EnKF-MCMC. The units are C${}_{\mathrm{dop}}\;$(cm${}^{3}$) and ρ (q/nm${}^{2}$).

**Figure 8 sensors-22-04785-f008:**
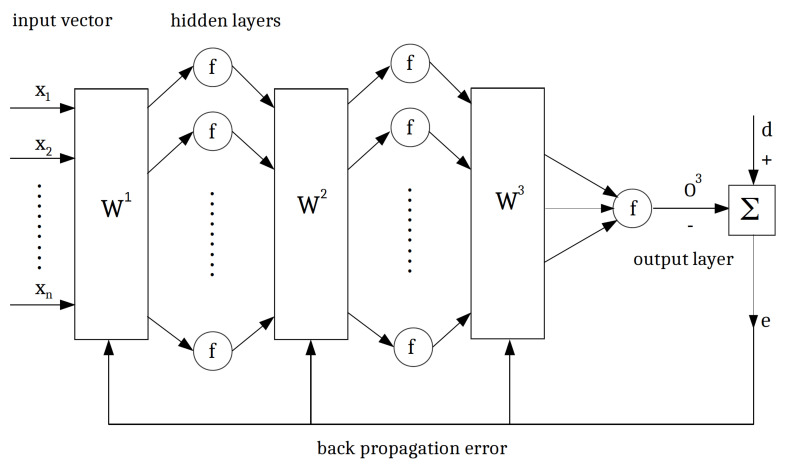
The structure of the MFNNs algorithm.

**Figure 9 sensors-22-04785-f009:**
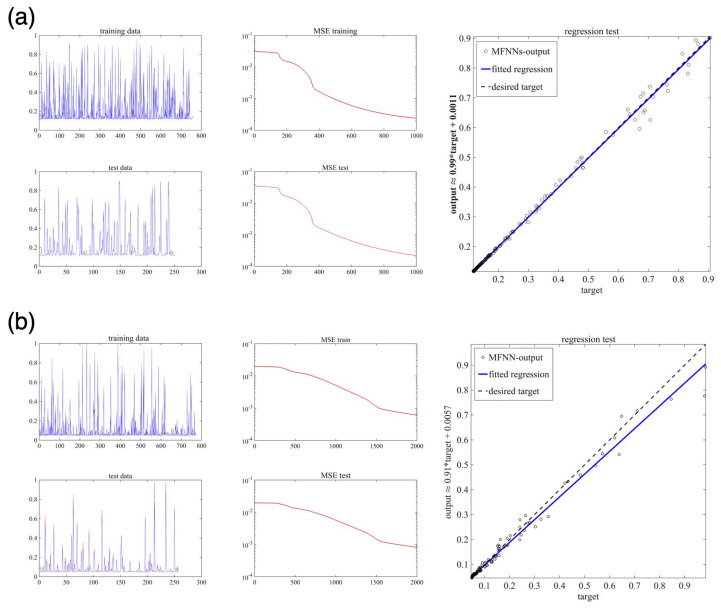
The performance of MFNNs algorithm for Case 1 (**a**), Case 2 (**b**), and Case 3 (**c**). In the first column, the desired trajectories (shown in blue) are compared with the MFNN output (shown in red). In the second column, we have the relative MSE, and the regression test is given in the third column.

**Figure 10 sensors-22-04785-f010:**
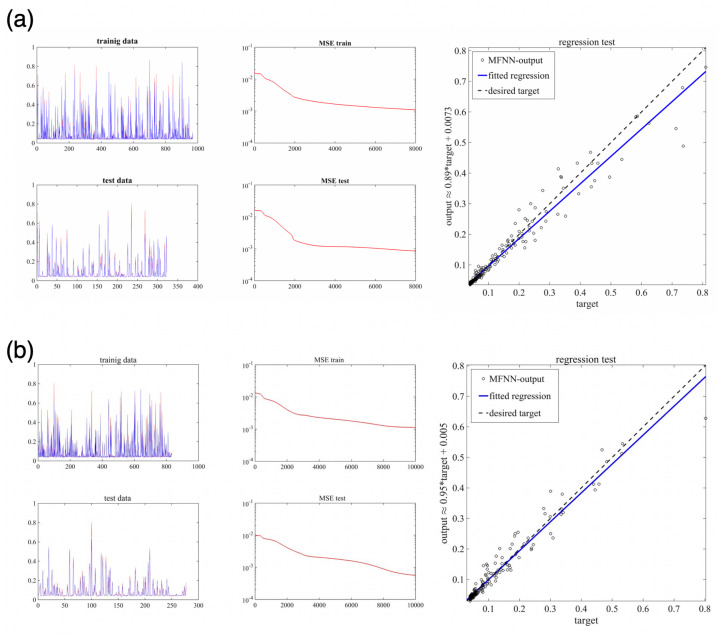
The performance of the MFNNs algorithm for Case 4 (**a**) and Case 5 (**b**). In the first column, the desired trajectories (shown in blue) are compared with the MFNN output (shown in red). In the second column, we have the relative MSE, and the regression test is given in the third column.

**Table 1 sensors-22-04785-t001:** The computational features and the results of the Bayesian inversion.

Parameter	Min	Max	EnKF (Median)	True Values	Acceptance Rate
C${}_{\mathrm{dop}}\;$(cm${}^{3}$)	1 $\times \;10^{15}$	5 $\times \;10^{16}$	9.4 $\times \;10^{15}$	1 $\times \;10^{16}$	91%
ρ (q/nm${}^{2}$)	−5	1	−1.55	−1.5	86%

**Table 2 sensors-22-04785-t002:** The range of parameters used to compute the electrical current in different cases.

Cases	Inputs	$V_{g}$ [V]	SiO${}_{2}$ [nm]	N${}_{W}$ [nm]	$C_{\mathrm{dop}}\;\left[{\mathrm{cm}}^{3}\right]$	N${}_{H}$ [nm]
Case 1	1	U(−1,−5)	8	100	1$\times \;10^{16}$	50
Case 2	2	U(−1,−5)	U(5,15)	100	1$\times \;10^{16}$	50
Case 3	3	U(−1,−5)	U(5,15)	U(80,120)	1$\times \;10^{16}$	50
Case 4	4	U(−1,−5)	U(5,15)	U(80,120)	$U(1\times 10^{15},5\times 10^{16})$	50
Case 5	5	U(−1,−5)	U(5,15)	U(80,120)	$U(1\times 10^{15},5\times 10^{16})$	U(40,60)

**Table 3 sensors-22-04785-t003:** The features of the MFNNs algorithm including the MSE of training and test processes.

Case	No. Neurons in 1st Hidden Layer	No. Neurons in 2nd Hidden Layer	MSE-Train	MSE-Test	No. Epochs	η
1	10	4	0.00057	0.00061	1000	0.1
2	20	7	0.00147	0.00184	2000	0.2
3	20	7	0.00181	0.000836	4000	0.2
4	20	7	0.000842	0.000517	8000	0.2
5	20	7	0.0011	0.000058	10,000	0.2

## Data Availability

Not applicable.

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
