# Peer review of "Rational Design of Field-Effect Sensors Using Partial Differential Equations, Bayesian Inversion, and Artificial Neural Networks"

_sensors, 2022, doi:10.3390/s22134785_

Round 1

Reviewer 1 Report

This manuscript describes the theoretical and computational aspects of forward and backward modeling of biosensitive sensorswhich is interesting and meaningful. But some questions and suggestions should be addresses:

1. The introduction can be more concise and logical.

2. Essay writing needs improvement.

3. In the method part, the principle flow chart can be used to introduce the connection and function of each part, making the structure of the article clearer.

4. Could you introduce the PROPKA algorithm, Scharfetter-Gummel Type Iteration, Delayed Rejection Adaptive Metropolis (DRAM), the Metropolis- Hastings algorithm and filtering algorithm in method in detail?

5. It is better to use alphabetic annotation to distinguish pictures in Fig6, Fig 8 and Fig9.

6. It seems unnecessary to use a three line meter to describe the summary of the algorithm in Algorithm 1.

7. The units of horizontal and vertical coordinates need to be marked out in Fig4, Fig6, Fig 8 and Fig9.

Author Response

The authors would like to thank the reviewers for their time to write the comments, remarks, and questions. In fact, the given details helped significantly us to improve the manuscript We have revised the manuscript with additional explanations as suggested by the reviewers and we believe that we have addressed all the points raised by the reviewer in this revision. The changes in the manuscript as well as our answers are marked in blue color.

Reviewer 1

This manuscript describes the theoretical and computational aspects of forward and backward modeling of biosen- sitive sensors, which is interesting and meaningful. But some questions and suggestions should be addresses:

  1. The introduction can be more concise and logical.

    In order to make the introduction more concise, we have shortened some of the sentences, please see pages 1-2.

  2. Essay writing needs improvement.

    We have reworded the introduction, please see pages 1-2.

  3. In the method part, the principle flow chart can be used to introduce the connection and function of each part, making the structure of the article clearer.
    We appreciate the reviewer for the very useful comment. We have added the flowchart, please see Figure 2 to explain how the Bayesian inversion can be used to identify the material parameters and the Scharfetter-Gummel iteration is employed to solve the coupled system of equations.

  4. Could you introduce the PROPKA algorithm, Scharfetter-Gummel Type Iteration, Delayed Rejection Adaptive Metropolis (DRAM), the Metropolis- Hastings algorithm and filtering algorithm in method in detail?
    We added the description of the PROPKA algorithm, please see pages 10-11. The implementation of the Scharfetter-Gummel (using a flowchart) is shown in Figure 2. The delayed rejection adaptive Metropolis (DRAM) has not been used in the paper. Actually, EnKF-MCMC shows a better performance compared to DRAM. A review paper considering the common Bayesian inversion method (also EnKF-MCMC and DRAM), including the relative MATLAB codes can be found in [24] (please see page 7). Regarding the Kalman filter, the details mentioned in given in Subsection 3.1. More technical details can be found in [24].

  5. It is better to use alphabetic annotation to distinguish pictures in Fig6, Fig 8 and Fig9.

    We have used the alphabetic annotation as requested by the reviewer.

  6. It seems unnecessary to use a three line meter to describe the summary of the algorithm in Algorithm 1.

    We slightly worked on the algorithm, please see Algorithm 1. If it needs more modification, we appreciate if

    the respected reviewer could inform us.

  7. The units of horizontal and vertical coordinates need to be marked out in Fig4, Fig6, Fig 8 and Fig9.

    We added the units to Figure 5 and Figure 7 (Fig 4 and Fig. 6 in the previous version). Regarding Figures 9-10 (Fig. 8 and Fig. 9), we used the normalized values which are unitless.

Reviewer 2 Report

The article with title : “Rational Design of Field-Effect Sensors using Partial Differential Equations, Bayesian Inversion, and Artificial Neural Networks”, deals with field-effect sensors, more precisely field-effect transistors and their response extracted by using a combination of theory and computational methods that include: i) partial differential equations to model the electrical behavior and ii) the Bayesian Marchov-Chain Monte-Carlo method to address concentrations of molecules.

The article is well written, and shows an interesting approach to study and understand the response of field-effect based sensors. In combination with experimental data, this is actually a very strong and convenient tool to progress in this multidisciplinary area.

I think the article is suitable for publication, I did spot a few minor issues though:

line 37: typo: molecules is written twice

line 84: beta := q/(kB*T) … in the article maybe better to change to: beta = q/(kB*T) ?

line 86: Gamma := Omega_Si Omega_liquid. It should be Omega_SiO2 I belive ?

line 148: add sentence to describe where the 2ZCH molecule is sitting and what does it do. I guess that upon the presence of prostate cancer this marker can respond changing its “charge” upon target binding. Please comment this in a sentence or two, because it will help also a much more broad readership to follow the work.

line 203: -5Vm change to -5 V

A part form these minor typos, I do have a question/comment that should be discussed brievly if possible:

The SiO2, is a very rich interface, and it is responsive to pH on its own even without functionalization. This is probably very difficult to address and goes beyond the scope of the current work, but from a sensing point of view (more particularly, selective sensing) the interface between the insulator and the liquid and the immobilized receptor molecules is crucial for the sensor. It would be great if the authors could comment if their approach allows also to include issues related to surface states and parasitic responses, i.e. a change in current other than binding a target molecule. For instance, would it be possible to predict the impact of molecular surface coverage on target binding / parasitic response ?

Author Response

The authors would like to thank the reviewers for their time to write the comments, remarks, and questions. In fact, the given details helped significantly us to improve the manuscript We have revised the manuscript with additional explanations as suggested by the reviewers and we believe that we have addressed all the points raised by the reviewer in this revision. The changes in the manuscript as well as our answers are marked in red color.

Reviewer 2

The article with title : “Rational Design of Field-Effect Sensors using Partial Differential Equations, Bayesian Inversion, and Artificial Neural Networks”, deals with field-effect sensors, more precisely field-effect transistors and their response extracted by using a combination of theory and computational methods that include: i) partial differ- ential equations to model the electrical behavior and ii) the Bayesian Marchov-Chain Monte-Carlo method to address concentrations of molecules.

The article is well written, and shows an interesting approach to study and understand the response of field-effect based sensors. In combination with experimental data, this is actually a very strong and convenient tool to progress in this multidisciplinary area.
I think the article is suitable for publication, I did spot a few minor issues though:

  1. line 37: typo: molecules is written twice

    We have corrected the introduction, please see page 2

  2. line 84: beta := q/(kB*T) . . . in the article maybe better to change to: beta = q/(kB*T) ? We have changed the notation, please see page 3.

  3. line 86: Gamma := Omega_S i ∩ Omega_liquid. It should be Omega_SiO2 I belive ? The reviewer is fully right regarding the interface, please see page see for Γ

  4. line 148: add sentence to describe where the 2ZCH molecule is sitting and what does it do. I guess that upon the presence of prostate cancer this marker can respond changing its “charge” upon target binding. Please comment this in a sentence or two, because it will help also a much more broad readership to follow the work.
    That is a very interesting comment. We added a couple of lines to explain the effect of charged molecules , please see Figure 11.

  5. line 203: -5Vm change to -5 V

    We corrected the wrong gate voltage, please see page 13

  6. A part form these minor typos, I do have a question/comment that should be discussed brievly if possible:
    The SiO2, is a very rich interface, and it is responsive to pH on its own even without functionalization. This is probably very difficult to address and goes beyond the scope of the current work, but from a sensing point of view (more particularly, selective sensing) the interface between the insulator and the liquid and the immobilized receptor molecules is crucial for the sensor. It would be great if the authors could comment if their approach 

    allows also to include issues related to surface states and parasitic responses, i.e. a change in current other than binding a target molecule. For instance, would it be possible to predict the impact of molecular surface coverage on target binding / parasitic response ?
    That is a very interesting question. The system of equations is capable of modeling the surface charges at the surface. In a previous work, we developed a Monte-Carlo approach to simulate the charges around a charged biomolecule at a charged surface [47]. Furthermore, in [48], a nonlinear Poisson model was used to calculate the free energies of various molecule orientations in dependence on the surface charge. Based on the free energies, the probabilities of the orientations were calculated and hence the biological noise was simulated.
